# Effective Circulating Tumor Cell Isolation Using Epithelial and Mesenchymal Markers in Prostate and Pancreatic Cancer Patients

**DOI:** 10.3390/cancers15102825

**Published:** 2023-05-18

**Authors:** Jiwon Cha, Hyungseok Cho, Jae-Seung Chung, Joon Seong Park, Ki-Ho Han

**Affiliations:** 1Department of Nanoscience and Engineering, Center for Nano Manufacturing, Inje University, Gimhae 50834, Republic of Korea; ckjw51@gmail.com (J.C.); elshaddai88@naver.com (H.C.); 2Department of Urology, Haeundae Paik Hospital, Inje University, Busan 48108, Republic of Korea; biogen@hanmail.net; 3Pancreatobiliary Cancer Clinic, Department of Surgery, Gangnam Severance Hospital, Yonsei University College of Medicine, Seoul 06229, Republic of Korea; jspark330@yuhs.ac

**Keywords:** CTCs, CTC-dChip, EpCAM, microfluidics, vimentin, epithelial and mesenchymal markers

## Abstract

**Simple Summary:**

Circulating tumor cells (CTCs) are important diagnostic and prognostic markers in cancer patients. However, current methods for isolating CTCs primarily focus on epithelial markers, such as EpCAM, leading to the underrepresentation of mesenchymal CTCs. This study proposes a novel method for continuous CTC isolation using magnetic nanoparticles coated with both EpCAM and vimentin antibodies, a mesenchymal marker. Using lateral magnetophoresis, the method demonstrated increased efficiency and reliability in isolating CTCs from patients with prostate (*n* = 17) and pancreatic (*n* = 5) cancer. Results indicated that using both EpCAM and vimentin antibodies significantly improved CTC isolation compared to using either marker alone, regardless of cancer stage.

**Abstract:**

Circulating tumor cells (CTCs) display antigenic heterogeneity between epithelial and mesenchymal phenotypes. However, most current CTC isolation methods rely on EpCAM (epithelial cell adhesion molecule) antibodies. This study introduces a more efficient CTC isolation technique utilizing both EpCAM and vimentin (mesenchymal cell marker) antibodies, alongside a lateral magnetophoretic microseparator. The effectiveness of this approach was assessed by isolating CTCs from prostate (*n* = 17) and pancreatic (*n* = 5) cancer patients using EpCAM alone, vimentin alone, and both antibodies together. Prostate cancer patients showed an average of 13.29, 11.13, and 27.95 CTCs/mL isolated using EpCAM alone, vimentin alone, and both antibodies, respectively. For pancreatic cancer patients, the averages were 1.50, 3.44, and 10.82 CTCs/mL with EpCAM alone, vimentin alone, and both antibodies, respectively. Combining antibodies more than doubled CTC isolation compared to single antibodies. Interestingly, EpCAM antibodies were more effective for localized prostate cancer, while vimentin antibodies excelled in metastatic prostate cancer isolation. Moreover, vimentin antibodies outperformed EpCAM antibodies for all pancreatic cancer patients. These results highlight that using both epithelial and mesenchymal antibodies with the lateral magnetophoretic microseparator significantly enhances CTC isolation efficiency, and that antibody choice may vary depending on cancer type and stage.

## 1. Introduction

Circulating tumor cells (CTCs) are cells that enter the bloodstream after detaching from tumor tissues. These cells contribute to metastasis by migrating from the primary tumor site to other organs and potentially lead to tumor recurrence through dormant micrometastases. As CTCs persist after therapy, their levels can be measured to predict progression-free survival rates and overall survival rates in cancer patients, showing a proportional relationship to disease progression when carcinoma recurs [1]. As a result, the quantity of CTCs in peripheral blood serves as a valuable diagnostic, prognostic, and treatment monitoring marker in cancer [2]. Some studies [3,4] have also shown that primary cancers begin shedding neoplastic cells into the circulation even at an early stage, suggesting that the number of CTCs may reflect the tumor burden throughout cancer progression. A particularly important advantage of CTC detection is its minimally invasive nature, allowing for frequent resting, whereas repeated invasive procedures like conventional tissue biopsies may result in limited patient compliance [5]. Furthermore, CTCs can provide crucial genetic information about cancer [6,7,8].

In the circulatory system, most CTCs undergo anoikis-induced cell death [9], with only a small fraction developing anoikis resistance through epithelial-to-mesenchymal transition (EMT) [10]. The surviving CTCs in the bloodstream exhibit a heterogeneous phenotype spectrum between epithelial to mesenchymal cells [11,12]. However, most current representative methods for CTC isolation, such as CellSearch [5], magnetic-activated cell sorting (MACS) [13], and MagSweeper [14], primarily rely on the epithelial cell adhesion molecule (EpCAM). Consequently, conventional EpCAM-based techniques struggle to effectively isolate mesenchymal CTCs, which display increased mesenchymal and decreased epithelial surface markers due to EMT. Recent studies have highlighted the significant role of mesenchymal CTCs in cancer metastasis [15,16] and their direct correlation with cancer treatment resistance [17,18,19]. Therefore, effective isolation of mesenchymal CTCs is crucial for advancing cancer genomics and treatment strategies. To effectively isolate CTCs across a wide phenotypic range, from epithelial to mesenchymal, a novel method incorporating both epithelial and mesenchymal markers is necessary.

A few recent studies have sought to enhance CTC isolation efficiency by simultaneously utilizing EpCAM and the mesenchymal marker N-cadherin. For instance, Wang et al. developed fluorescent magnetic nanoparticles coated with both EpCAM and N-cadherin antibodies, reporting CTC isolation results for 10 breast cancer patients using the conventional MACS method [20]. Po et al. created magnetic nanoparticles coated separately with EpCAM and N-cadherin antibodies, reporting CTC isolation results for 18 ovarian cancer patients using a commercial immunomagnetic capture system (Isoflux, Fluxion) [21]. These results indicated that, compared to CTC isolation efficiency with magnetic nanoparticles coated only with EpCAM antibodies, their efficiency using only N-cadherin antibodies was over twice as high, and their efficiency using both EpCAM and N-cadherin antibodies was more than three times higher. Despite these promising findings, efforts to effectively isolate CTCs using both epithelial and mesenchymal markers remain in the early stages. Most current studies [22,23] have been limited to using only EpCAM and N-cadherin antibodies. Furthermore, previous CTC isolation research has largely depended on immunoaffinity or immunomagnetism-based CTC capture techniques, which necessitate a retrieval process for collecting captured CTCs and may compromise overall separation efficiency due to potential CTC loss.

In this study, we propose a novel method for continuous CTC isolation using magnetic nanobeads coated with both EpCAM and vimentin antibodies, epithelial and mesenchymal markers, respectively. This approach employs lateral magnetophoresis [24,25] to achieve high separation efficiency and reliability. Vimentin is an ideal marker for isolating mesenchymal CTCs, as it is overexpressed during EMT [26,27]. Studies [28,29] have shown that vimentin antibodies are more effective for isolating CTCs from certain cancers, such as breast and pancreatic cancer, than EpCAM antibodies alone. Additionally, previous research [30] has demonstrated that cell surface vimentin is a useful biomarker to isolate CTCs in most solid cancers. In this study, we isolated CTCs from prostate (*n* = 17) and pancreatic (*n* = 5) cancer patients using a disposable lateral magnetophoretic microseparator, comparing the isolated CTC count, white blood cell (WBC) depletion rate, and CTC purity based on the EpCAM and vimentin antibodies combination. Our results demonstrate the effectiveness of using both antibodies for CTC isolation and offer insight into the performance of the two CTC markers for isolation.

## 2. Materials and Methods

### 2.1. Working Principle

Sample preparation processes involved mixing blood with either EpCAM antibodies, anti-vimentin antibodies, or both types of magnetic nanobeads, as shown in Figure 1a. These nanobeads specifically bind to epithelial and mesenchymal CTCs, enabling their tagging and subsequent isolation. The CTC-dChip, which utilizes lateral magnetophoresis to isolate magnetic particles, is designed with two inlets for introducing the sample and buffer, and two outlets for discarding normal blood cells and collecting CTCs, as illustrated in Figure 1b. The microchannel is equipped with a ferromagnetic wire array positioned at an angle (*θ* = 5.7°) related to the direction of fluid flow, which generates a high-gradient magnetic field when an external magnetic field is applied. Consequently, CTCs bound with magnetic nanobeads experience both magnetic force (*F_m_*) and hydrodynamic drag force (*F_d_*), which leads to the generation of a lateral magnetic force (*F_l_*) that directs them towards the CTC outlet (Figure 1b). The sample and phosphate-buffered saline (PBS) containing 0.2% bovine serum albumin (BSA) are introduced through the respective sample and buffer inlets. CTCs gathered from the CTC outlet were identified using confocal immunofluorescence analysis (Figure 1c) and subsequently quantified.

### 2.2. Instrument Setup

The CTC-dChip is composed of a disposable microchannel and a reusable magnetic gradient substrate incorporating an embedded ferromagnetic wire array. The disposable microchannel is created by bonding a microchannel-structured polydimethylsiloxane (PDMS) replica with a 12-μm-thick silicone-coated release polyethylene terephthalate (PET) film. The fabrication processes of both the disposable microchannel and the reusable magnetic gradient substrate are detailed in Appendix A. Figure 2a,b display the fabricated disposable microchannel and reusable magnetic gradient substrate, which can be effortlessly assembled or dissembled using vacuum pressure (Figure 2c).

For the CTC-dChip instrument setup, the reusable substrate is positioned at the center of two stacked permanent magnets. Upon aligning the disposable microchannel with the reusable substrate, an air vacuum pressure of −80 kPa is exerted via the vacuum hole to securely affix the microchannel and substrate (Figure 2d). This vacuum connection firmly fastens them together, eliminating any air gap between the microchannel and substrate. As a result, the magnetic field produced by the ferromagnetic wires can successfully permeate the 12-μm PET film, influencing the movement of CTCs bound with magnetic nanobeads as they pass through the microchannel. Since the disposable microchannel is constructed from polymeric materials such as PDMS and PET film, the fabrication cost remains low. Consequently, the disposable microchannel can be replaced with a new one for each experiment, making the CTC-dChip a safe and cost-effective option that prevents biological contamination.

### 2.3. Sample Preparation

Healthy human blood was gathered using Vacutainer tubes with the anticoagulant EDTA and processed within 6 h. For analytical assessment, PANC-1 and LNCaP cancer cell lines were marked using a green fluorescent dye (SYTO 13, Invitrogen) to visualize the nuclei and then cautiously suspended to achieve a concentration of approximately 10^4^ cells per milliliter. Next, 20 μL of the concentrated mixture was added to 3 mL of healthy whole blood in a 15 mL conical tube, resulting in approximately 200 spiked cell lines. Blood samples were collected from five healthy donors, a patient with benign prostatic hyperplasia, and patients with prostate cancer (*n* = 17) and pancreatic cancer (*n* = 5). Detailed information about the blood donors can be found in Appendix A.

To isolate nucleated cells in prepared blood sample (1.5 to 3 mL), a 1.119 g/mL density Histopaque medium (Sigma-Aldrich, St. Louis, MO, USA) was used for density gradient centrifugation at 700× *g* for 30 min. The isolated cells were then transferred into a 30 mL conical tube containing 5 mL of ice-cold PBS (0.2% BSA). After cleaning via centrifugation (200× *g*, 10 min), the cells were reconstituted in 1 mL of PBS (0.2% BSA) within a 1.5 mL microcentrifuge tube. After a final wash, the nucleated cells were suspended in 200 μL of ice-cold PBS with 0.2% BSA and mixed with anti-EpCAM (EasySep Human EpCAM Positive selection cocktail II, STEMCELL Technologies, Vancouver, BC, Canada) and biotinylated anti-vimentin (Vimentin Monoclonal Antibody (V9), Biotin, Invitrogen, Waltham, MA, USA) antibodies, followed by incubation on ice for 40 min. Streptavidin-anti-dextran (EasySep Biotin Selection cocktail, STEMCELL Technologies, Vancouver, BC, Canada) antibodies were added and incubated on ice for another 40 min. Magnetic nanobeads (EasySep Dextran RapidSpheres 50100, STEMCELL Technologies, Vancouver, BC, Canada) were then added according to the manufacturer’s instructions and incubated on ice for 60 min. To prepare the sample for a CTC-dChip, 800 μL of ice-cold PBS (0.2% BSA) was added to the collected cells, resulting in a 4-fold dilution. The cells collected from patient blood using CTC-dChips were fixed with 100 μL of 4% paraformaldehyde for 5 min, followed by permeabilization with 100 μL of 0.2% Triton X-100 for 5 min. Afterwards, the cells were stained for 30 min with a nucleic acid fluorescent dye (DAPI, Invitrogen, Waltham, MA, USA) to visualize the nuclei, anti-cytokeratin-Alexa 488 (eBioscience, Santa Clara, CA, USA) antibodies to detect CTCs, and anti-CD45-Alexa 647 (Biolegend, San Diego, CA, USA) antibodies to identify normal blood cells. CTCs were detected and counted with the aid of a confocal microscope (LSM510META; Carl Zeiss, Oberkochen, Germany).

To isolate 200 PANC-1 cells spiked into 2 mL healthy whole blood using MACS, the sample was diluted to 2.5 mL with ice-cold PBS containing 0.2% BSA in a 5 mL culture tube (LK LabKorea, Namyangju-si, Republic of Korea). Next, the tube was placed in a magnet for 5 min to allow for magnetic separation. The cell pellets were then resuspended in ice-cold PBS (0.2% BSA) after removal of the supernatant. This isolation procedure was performed three times, and then the tube was taken out of the magnet. The resulting cell pellets were suspended in 300 μL of ice-cold PBS (0.2% BSA). The isolated PANC-1 cells were identified using a fluorescent microscope, while WBCs were counted using a hemocytometer.

## 3. Results and Discussion

### 3.1. Analytical Evaluation Using Healthy Blood Spiked with Cancer Cell Lines

The performance of the CTC-dChips in isolating PANC-1 and LNCaP cancer cell lines using EpCAM and vimentin antibodies was quantitatively evaluated. PANC-1 cells, which have low surface expression levels of both EpCAM [31] and vimentin [32], showed an average recovery rate of 20% and 5% when using EpCAM and vimentin antibodies alone, respectively (Figure 3a). However, the recovery rate dramatically increased to about 50% when both EpCAM and vimentin antibodies were used, indicating the effectiveness of the CTC-dChips in isolating cells with low expression levels of these markers. In contrast, the recovery rate of LNCaP cells, which have high surface expression levels of EpCAM [33] and low surface expression levels of vimentin [26], was almost 100% when using EpCAM antibodies, but only averaged 5% when using vimentin antibodies.

The number of contaminating WBCs was approximately 200 per milliliter of blood, regardless of the antibodies used for isolation, as shown in Figure 3b. This result may be attributed to the unique isolation mechanism of the CTC-dChips and the quality of blood, independent of the antibodies used. The WBC depletion rate of CTC-dChips using healthy blood was calculated to be 4.4-log, based on the assumption that there are 5 × 10^6^ WBCs per milliliter of whole blood. The performance compares favorably with previous studies [34,35] reporting great results. Since the number of contaminating WBCs is nearly constant, the purity of cancer cells retrieved from spiked blood samples increases with the recovery rate, as shown in Figure 3c.

Previous studies [36,37] have primarily utilized the MACS method for isolating CTCs. In this study, the CTC-dChip was used to isolate PANC-1 and LNCaP cancer cells spiked into healthy whole blood, and the results were compared with those obtained using MACS to access the performance of the CTC-dChip. The high isolation efficiency of the CTC-dChip enables the retrieval of PANC-1 cells with a recovery rate of approximately 60% and LNCaP cells with a recovery rate of nearly 100% using both EpCAM and vimentin antibodies, consistent with the results shown in Figure 3a. Interestingly, the recovery rates of MACS for both PANC-1 and LNCaP cells were found to be approximately 60% compared to that of the CTC-dChip, as demonstrated in Figure 4a. This suggests that the MACS method loses around 40% of the captured target cells during the isolation process, a finding that is consistent with previous studies [25,38]. As demonstrated by the recovery rate of the CTC-dChip, the MACS method exhibited the highest recovery rate when using both EpCAM and vimentin antibodies, followed by using EpCAM and vimentin antibodies individually. Furthermore, the number of contaminating WBCs with the CTC-dChips was at least 500-fold smaller than that obtained with MACS (Figure 4b), resulting in the purity of PANC-1 and LNCaP cells retrieved by the CTC-dChips being at least 800-fold higher than that achieved by MACS (Figure 4c). Consequently, the high WBC depletion rate allows for the collection of highly pure CTCs from the blood of cancer patients, facilitating highly precise genetic analysis using the isolated CTCs.

The viability of cells isolated using the CTC-dChips were evaluated using a fluorescence-based viability assay (EZ-View Live/Dead Cell staining Kit, BIOMAX). The viability of LNCaP cells before isolation was measured at 100%. After isolation with EpCAM antibodies alone, vimentin antibodies alone, or both combined, the viability of LNCaP cells was observed to be 99.5% (the number of live/dead cells: 195/1), 100% (8/0), and 99.0% (197/2), respectively, as shown in Appendix A. These results indicate that the viability of cells isolated by the CTC-dChips are not significantly affected, suggesting that CTCs isolated using the CTC-dChips can be utilized for subsequent cellular-level analyses, such as culture, drug screening, and patient-derived xenografts.

### 3.2. Clinical Evaluation Using Blood of Patients with Prostate and Pancreatic Cancer

To evaluate the effectiveness of the proposed CTC isolation method that utilizes both EpCAM and vimentin antibodies, we performed experiments on blood samples obtained from five healthy donors, a patient with benign prostatic hyperplasia, and patients diagnosed with prostate cancer (*n* = 17) and pancreatic cancer (*n* = 5). We collected three 3-mL blood samples from each patient and used EpCAM antibodies alone, vimentin antibodies alone, and both of them together to isolate CTCs.

#### 3.2.1. Prostate Cancer Patients

As a result of the analytical evaluation using PANC-1 and LNCaP cells, the efficiency of isolating CTCs from the blood of patents with prostate cancer was found to be most effective using both EpCAM and vimentin antibodies, as demonstrated in Figure 5. Based on the results from five healthy donors and a patient with benign prostatic hyperplasia, the lower limit of detection can be set to 1 CTC/mL when both EpCAM and vimentin antibodies are used. Consequently, in this study, the CTC detection rate was 100% for patents with prostate cancer (*n* = 17). Using EpCAM antibodies alone, an average of 13.29 CTCs/mL (range, 3 to 48) was found in the blood samples from prostate cancer patients. When using vimentin antibodies alone, we found an average of 11.13 CTCs/mL (range, 1 to 56). However, when using EpCAM and vimentin antibodies together, we found an average of 27.95 CTCs/mL (range, 5 to 122), which was two and two and a half times higher than those isolated using EpCAM and vimentin antibodies alone, respectively (Figure 5a). We also found that the number of CTCs isolated using EpCAM antibodies alone was generally higher than those using vimentin antibodies alone in patients with localized prostate cancer (T2-T4), except for one patient. However, in eight out of eleven patients with metastatic hormone-sensitive prostate cancer (mHSPC) and metastatic castration-resistant prostate cancer (mCRPC), the number of CTCs isolated using vimentin antibodies was higher than or equal to those using EpCAM antibodies. This supports previous studies [11,39,40] reporting a high proportion of mesenchymal CTCs in advanced cancer. Appendix A displays the CTCs that were isolated from patients with prostate cancer.

The number of contaminating WBCs was found to be constant, ranging from 1000 to 3000 per milliliter (mean, 1816 WBCs/mL of whole blood) regardless of the combination of EpCAM and vimentin antibodies (Figure 5b). The WBC depletion rate was calculated to be an average of 3.44-log, which is 10-fold lower compared to the results obtained using healthy blood spiked with cancer cell lines. We believe that this is due to the poor condition of the blood from cancer patients. The purity of CTCs was generally proportional to the number of isolated CTCs, with high values ranging from 0.65% to 4.48% in metastatic cancer (Figure 5c). When using both EpCAM and vimentin antibodies, the average purity was 2.17%. These findings demonstrate the effectiveness of the proposed CTC isolation method for prostate cancer patients and provide valuable information for future research on CTCs.

#### 3.2.2. Pancreatic Cancer Patients

Compared to other types of cancer, pancreatic cancer has been shown to have the lowest number of CTCs isolated by using EpCAM antibodies [5]. This may be due to pancreatic tumor cells being in the EMT states from the early stages of cancer [41]. To evaluate the usefulness in isolating CTCs in pancreatic cancer using vimentin antibodies, CTCs from five patients with pancreatic cancer were isolated using EpCAM antibodies alone, vimentin antibodies alone, and both of them together. The average number of isolated CTCs using only EpCAM antibodies was 1.50 CTCs/mL (range, 0 to 3.5). The results using only vimentin antibodies showed that the average number of isolated CTCs was 3.44 CTCs/mL (range, 2 to 6). When using both EpCAM and vimentin antibodies, the average number of isolated CTCs was 10.82 CTCs/mL (range, 2.7 to 22.4), as shown in Figure 6a. This value is roughly 40-fold higher than the results (0.27 [5] and 0.18 CTCs/mL [42]) of previous works using the CellSearch^®^ system (Menarini Silicon Biosystems, Bologna, Italy), which is based on EpCAM antibodies, to isolate CTCs from patients with pancreatic cancer. The number of CTCs isolated using vimentin antibodies was more than double that isolated using EpCAM antibodies, consistent with previous findings [28,30] that demonstrate a higher capture rate of CTCs with mesenchymal cell markers compared to EpCAM antibodies in patients with pancreatic cancer. However, some studies [43,44] have reported a higher identification rate of epithelial CTCs relative to mesenchymal CTCs when filtered by ISET technology and analyzed using immunofluorescence images. Mechanical filtering methods such as ISET technology face challenges when isolating small-sized CTCs. In particular, mesenchymal CTCs, which are smaller [45] and more deformable [46,47,48] than epithelial CTCs, may be harder to isolate using these methods. This suggests that the isolation and identification of epithelial and mesenchymal CTCs could produce different results. It is essential to recognize that most CTCs exhibit an intermediate phenotype between epithelial and mesenchymal cells, and the identification of these subpopulations is heavily influenced by the criteria established for immunofluorescence intensity.

Similar to prostate cancer, CTCs were most effectively isolated in pancreatic cancer using both EpCAM and vimentin antibodies, and the number of isolated CTCs was seven and threefold higher than those using EpCAM and vimentin antibodies alone, respectively. The CTCs isolated from patients with pancreatic cancer are depicted in Appendix A. The number of contaminating WBCs ranged from 218 to 4575 per milliliter of blood (mean, 1570 WBCs/mL; WBC depletion rate, 3.5-log), possibly due to the worsening condition of blood in pancreatic cancer during transportation from a distant hospital for more than 8 h. Nevertheless, the results show that the number of contaminating WBCs does not depend on the combination of EpCAM and vimentin antibodies (Figure 6b). The purity of isolated CTC ranged widely from 0 to 9.32% (Figure 6c), and CTCs were still most effectively isolated when using both EpCAM and vimentin antibodies together.

## 4. Discussion

In this paper, we evaluated the efficacy of isolating CTCs from the blood of patients with prostate and pancreatic cancer using EpCAM (an epithelial marker) and vimentin (a mesenchymal marker) antibodies individually and in combination. Our findings showed that EpCAM antibodies were more effective in isolating CTCs from localized prostate cancer, whereas vimentin antibodies were more effective in advanced prostate cancer. Additionally, regardless of prostate cancer stage, the combination of EpCAM and vimentin antibodies proved to be more effective in isolating CTCs than using either EpCAM or vimentin antibodies alone. However, due to the small sample number of this study (*n* = 17 for prostate cancer and *n* = 5 for pancreatic cancer), we were unable to provide a comprehensive understanding of which antibodies are most useful for isolating CTCs based on cancer type and stage or progression of treatment. Nevertheless, we found that the combination of epithelial and mesenchymal antibodies was the most effective method for isolating CTCs in all enrolled patients with prostate and pancreatic cancer.

It should be noted that there are other epithelial markers, such as E-cadherin, and mesenchymal markers, such as N-cadherin and fibronectin, that could be utilized in combination with EpCAM and vimentin antibodies. Further analysis is required to determine which combination of epithelial and mesenchymal antibodies is most effective in isolating CTCs based on cancer type and stage. Our experimental results demonstrate that the proposed CTC-dChips can be used to isolate epithelial and mesenchymal CTCs individually or in combination with high efficiency and purity. This can be achieved by simply changing the antibody, making it suitable for use in general biological or chemical laboratories.

One limitation of this study is that we did not perform genetic analysis on the isolated CTCs in the small number of clinical samples analyzed. Therefore, while our findings suggest that the proposed method using the CTC-dChips is effective in isolating CTCs, we could not confirm the malignant nature of the isolated CTCs or address the heterogeneity among patient samples. To provide insights into specific genetic alterations and mutations unique to a cancer patient, which can aid in understanding the cancer biology and guide personalized treatment decisions, it is essential to perform genetic analysis on epithelial and mesenchymal CTCs. Obtaining genetic profiles from isolated CTCs is critical to demonstrate the superiority of the proposed isolation technique over established methods. In future studies, we plan to perform genetic analyses on epithelial and mesenchymal CTCs to identify significant biomarkers for predicting cancer prognoses and therapeutic responses, with the aim of achieving personalized precision cancer therapy. Furthermore, we plan to measure the number and portion of epithelial and mesenchymal CTCs isolated using EpCAM and vimentin antibodies together and extensively follow-up on patients with this data.

## 5. Conclusions

In summary, this study revealed that EpCAM antibodies were more effective in isolating CTCs from patients with localized prostate cancer, while vimentin antibodies were superior for in metastatic prostate cancer. Moreover, vimentin antibodies outperformed EpCAM antibodies in isolating CTCs from all pancreatic cancer patients. The results also emphasized that utilizing both epithelial (EpCAM) and mesenchymal (vimentin) antibodies with the lateral magnetophoretic microseparator significantly enhanced CTC isolation efficiency for both prostate and pancreatic cancer patients.

## Figures and Tables

**Figure 1 cancers-15-02825-f001:**
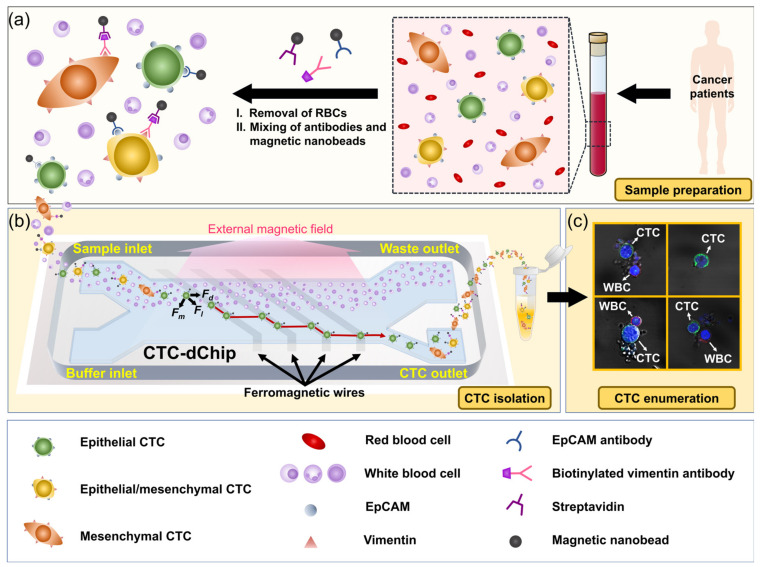
Workflow of CTC-dChip for isolating and identifying heterogeneous CTCs. (**a**) Schematic of sample preparation process, where anti-EpCAM and anti-vimentin magnetic nanobeads tag epithelial and mesenchymal CTCs in blood, respectively. (**b**) The blood sample and PBS solution are respectively injected through the sample and buffer inlets of the CTC-dChip. A uniform external magnetic field creates high-gradient magnetic fields around the ferromagnetic wires underlying the entire microchannel. CTCs that bind to magnetic nanobeads experience lateral movement and proceed to the CTC outlet, whereas WBCs are directed towards the waste outlet. (**c**) Isolated CTCs are identified and enumerated using immunofluorescence analysis with a confocal microscope.

**Figure 2 cancers-15-02825-f002:**
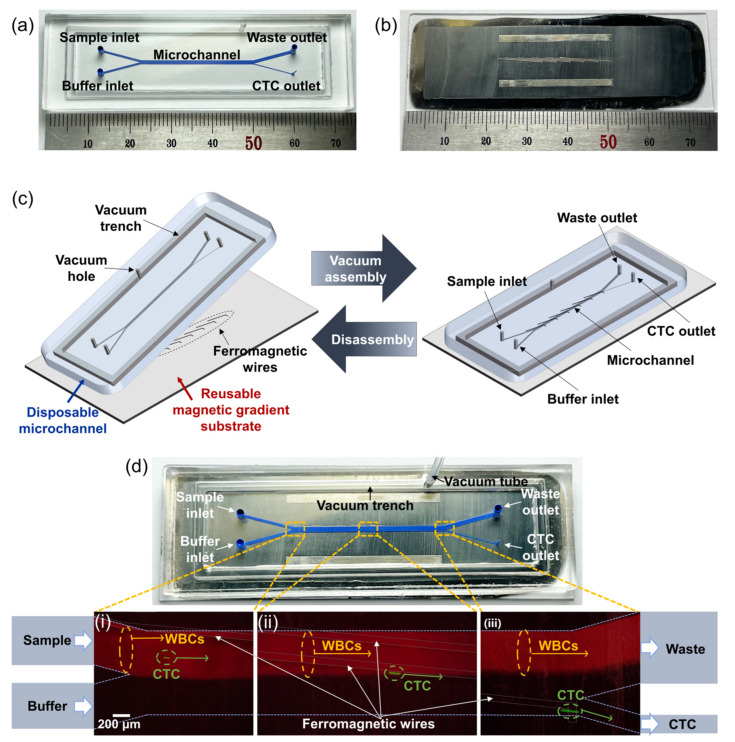
Photographs of the components of the CTC-dChip, including (**a**) the disposable microchannel and (**b**) the reusable magnetic gradient substrate with an inlaid ferromagnetic wire array. (**c**) Utilizing vacuum pressure allows for the effortless assembly or disassembly of the disposable microchannel and the reusable magnetic gradient substrate. (**d**) The vacuum assembled CTC-dChip is positioned atop two layered permanent magnets. (i) A mixture of WBCs (red) and spiked LNCaP prostate cancer cell lines (green) is combined with both anti-EpCAM and anti-vimentin magnetic nanobeads. PBS solution and the mixture are introduced through the buffer and sample inlets, respectively. (ii) Under an external magnetic field, LNCaP cells attached to magnetic nanobeads experience lateral movement, and (iii) subsequently travel to the CTC outlet, whereas WBCs are directed towards the waste outlet.

**Figure 3 cancers-15-02825-f003:**
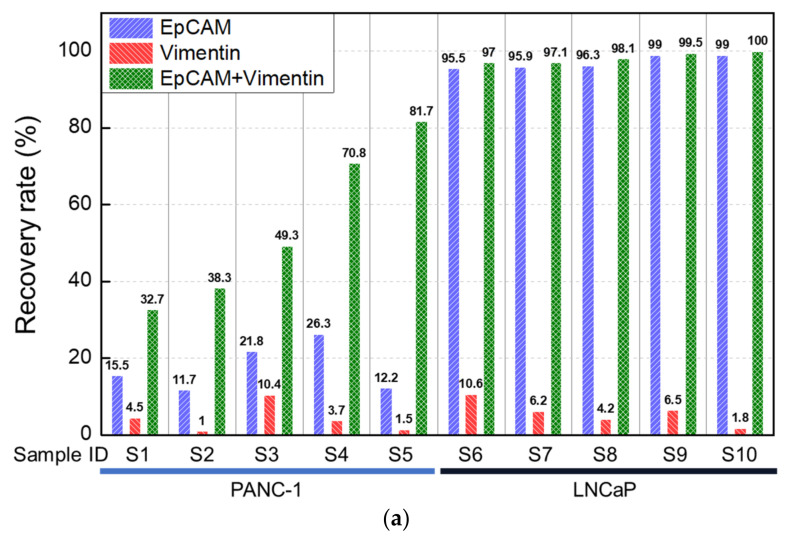
Analytical evaluation of the efficacy of CTC isolation using healthy blood spiked with PANC-1 or LNCaP cancer cell lines. (**a**) Recovery rates of the spiked PANC-1 and LNCaP cell lines, (**b**) the number of contaminating WBCs, and (**c**) purity of isolated PANC-1 and LNCaP cell lines using CTC-dChip with the combination of EpCAM and vimentin antibodies. Prostate (200 LNCaP) or pancreatic (200 PANC-1) cancer cell lines were spiked into 3 mL of healthy blood and labeled with anti-EpCAM magnetic nanobeads alone, anti-vimentin magnetic nanobeads alone, and anti-EpCAM and anti-vimentin magnetic nanobeads together, respectively, and isolated at sample and buffer flow rates of 2 mL/h.

**Figure 4 cancers-15-02825-f004:**
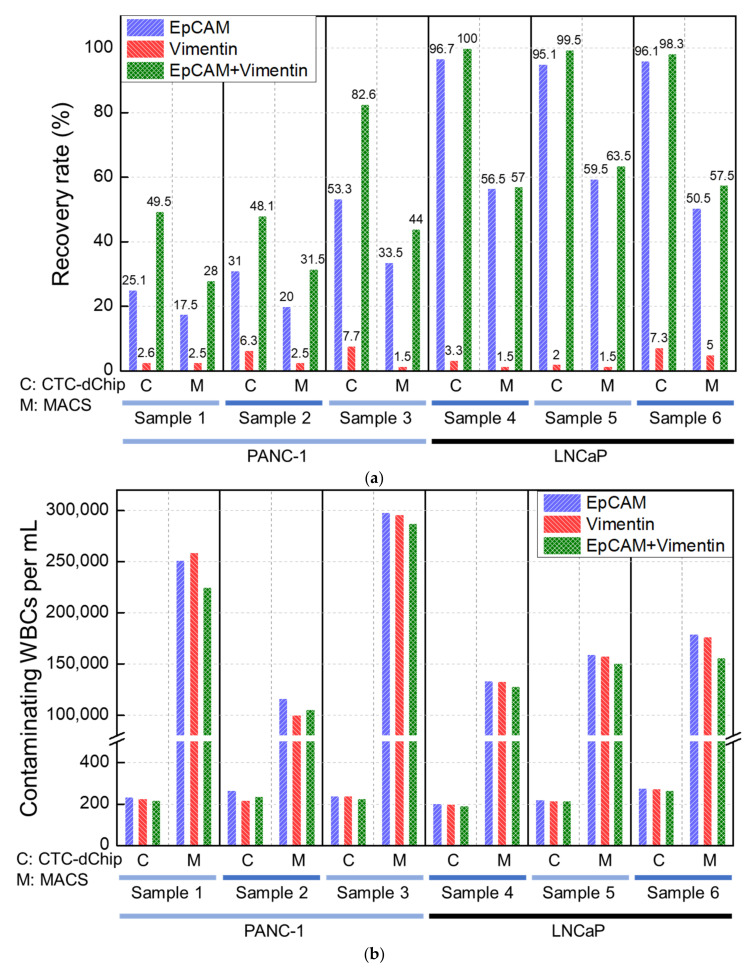
Comparison of the performance of CTC-dChip and MACS for isolating prostate (200 LNCaP) or pancreatic (200 PANC-1) cancer cell lines spiked into 2 mL of healthy blood. (**a**) Recovery rate for spiked cells and (**b**) the number of contaminating WBCs, isolated per milliliter of spiked blood. (**c**) Purity of the isolated PANC-1 and LNCaP cells.

**Figure 5 cancers-15-02825-f005:**
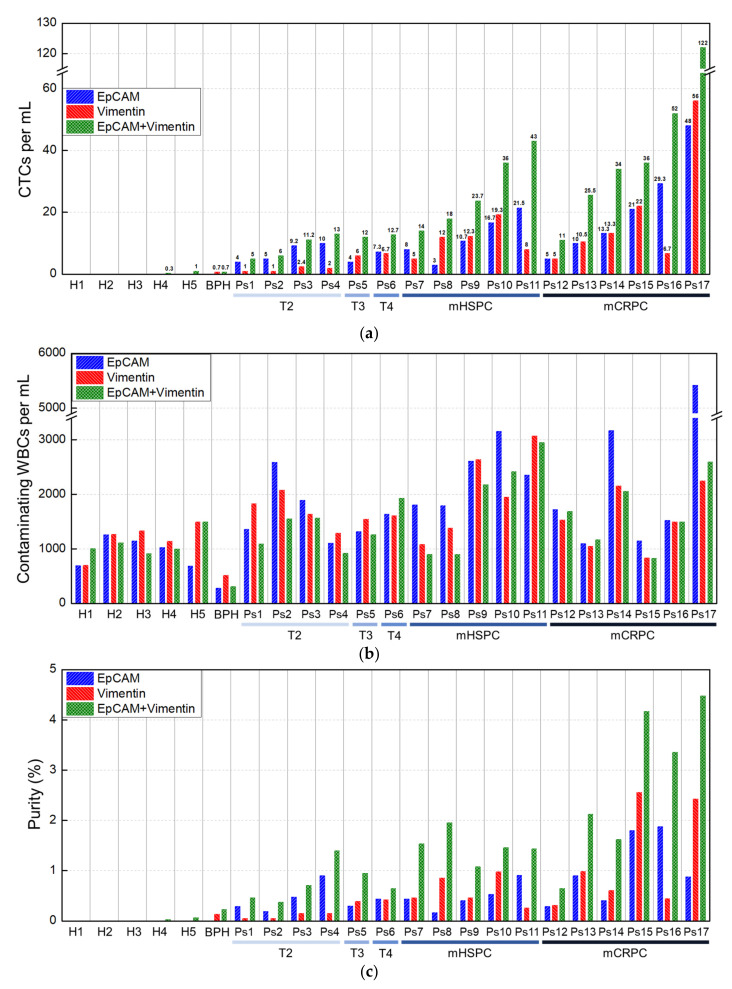
Clinical evaluation of blood from patients with prostate cancer. (**a**) Number of CTCs and (**b**) contaminating WBCs isolated per milliliter of blood from five healthy donors, a patient with benign prostatic hyperplasia (BPH), and patients (*n* = 17) with prostate cancer. (**c**) Purity of prostate CTCs.

**Figure 6 cancers-15-02825-f006:**
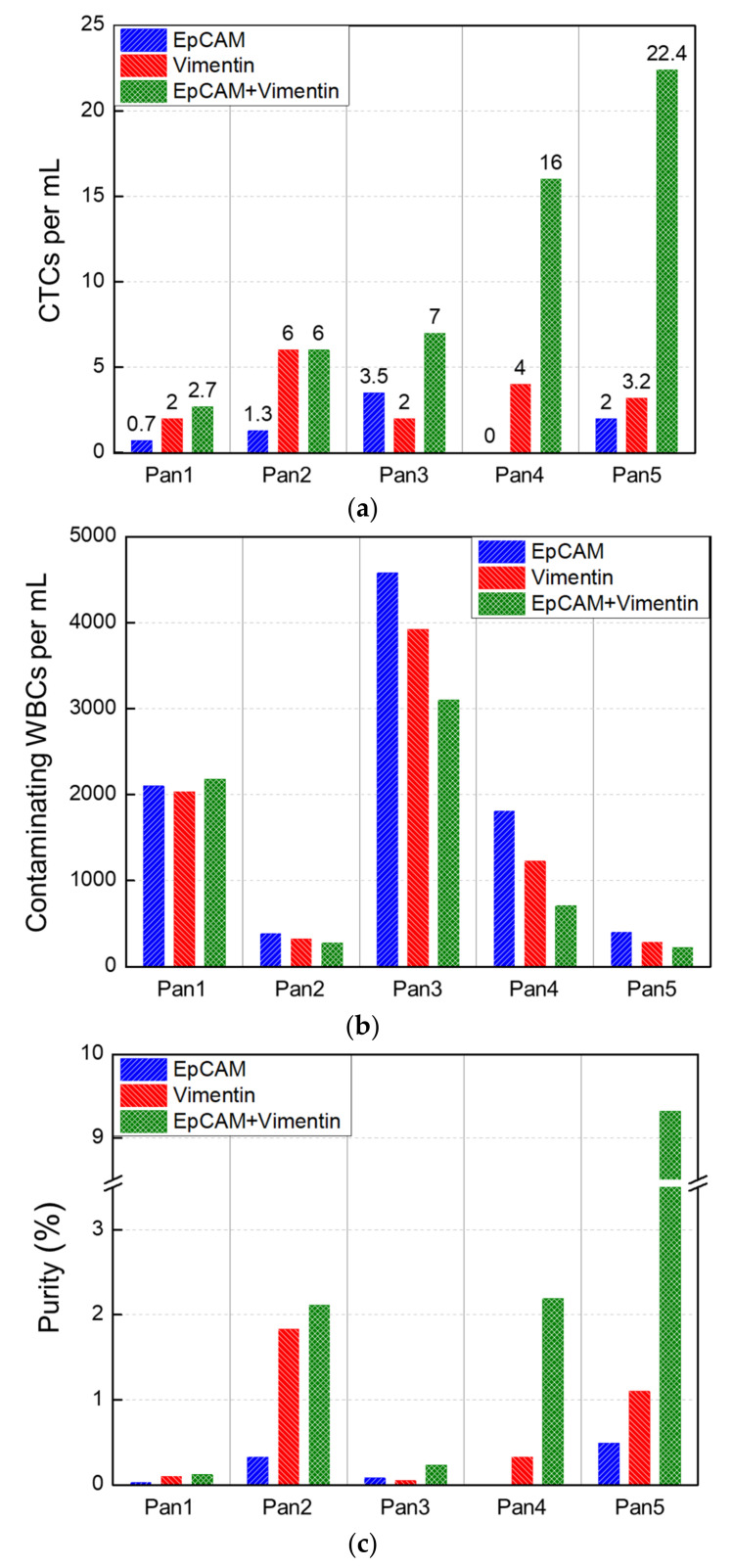
Clinical evaluation of blood from patients with pancreatic cancer. (**a**) Number of CTCs and (**b**) contaminating WBCs isolated per milliliter of blood from patients (*n* = 5) with pancreatic cancer. (**c**) Purity of pancreatic CTCs.

## Data Availability

Further information on the data collected are available from the corresponding author (K.-H.H.) upon reasonable request.

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
