# Peer review of "Effective Circulating Tumor Cell Isolation Using Epithelial and Mesenchymal Markers in Prostate and Pancreatic Cancer Patients"

_cancers, 2023, doi:10.3390/cancers15102825_

Round 1

Reviewer 1 Report

In their manuscript titled " Effective Circulating Tumor Cell Isolation Using Epithelial and Mesenchymal Markers in Prostate and Pancreatic Cancer Patients" written by Cha and colleagues, the authors investigated a new approach for isolating circulating tumor cells (CTCs) from the blood of prostate (n=17) and pancreatic cancer patients (n=5). In their limited cohort, they found that using a combination of EpCAM and vimentin antibodies was superior to using only one or the other in terms of CTC recovery. However, there are some major questions that need to be addressed before this manuscript can be published.

Firstly, the authors demonstrated that the combination of EpCAM and vimentin yields significantly higher numbers of CTCs than the sum of the EpCAM and vimentin groups (see fig 3). The authors should elaborate on this and provide more information on the surface expression of vimentin in PANC-1 and LNCaP cells. The use of a different prostate cell line expressing less EpCAM and more vimentin could be used to test the utility of the authors' system.

Secondly, it would be informative to include a comparison of the authors' method to the MACS protocol for the prostate cancer cell line, as was done for the pancreatic cell line (fig 4).

Thirdly, there are significant discrepancies between the number of CTCs reported in this manuscript and in a previously published paper (PMID: 30080739) that also investigated mixed epithelial/mesenchymal CTCs (pan-cytokeratin and vimentin) in a large pancreatic cancer patient cohort (n>130), as well as in another by the same group reporting on mixed CTCs prior to clinical recurrence (PMID: 36111892). The authors could speculate on the reason for these discrepancies, taking into account the different isolation methods used.

Fourthly, to demonstrate the reliability of the authors' method in a clinical setting, genetic verification of the detected CTCs could be performed, such as the identification of common driver mutations (i.e. KRAS for pancreatic cancer) in at least some of the cells.

Fifthly, concise clinical information regarding the stage of disease should be given in table S1, with the uniform use of the UICC stage advised.

Finally, it is significant that EpCAM + vimentin positive CTCs were detected in non-cancerous blood (see table S1 and fig 5), and the authors should discuss this finding and include additional healthy blood donors to determine a threshold for significant CTC detection.

In conclusion, while the presented study is very interesting and well written, some important questions need to be addressed before it can be published.

Reviewer 2 Report

The current manuscript describes a novel method for isolation of CTC in two different cancer- pancreatic and prostate cancer. The authors have presented their findings and methods in a clear descriptive manner. Unless already mentioned in the manuscript, it is vital to discuss the viability of the cells isolated and compare it with the existing methods for example MACS. I recommend the manuscript be accepted in the current format.

Reviewer 3 Report

Summary:

Circulating tumor cells (CTCs) are the cells that detached from primary tumor or secondary metastatic tumor and reached peripheral circulation. As CTC are vital for tumor spread, they are considered a highly attractive prognostic and predictive biomarker and a measure of treatment outcomes. A few recent studies have successfully enhanced CTC isolation efficiency by simultaneously utilizing epithelial marker EpCAM and the mesenchymal marker N-cadherin. However, the overall efficiency of CTCs separation still calls for more efforts. In this manuscript, Jiwon Cha, et al proposed a novel and more effective approach for CTC isolation using magnetic nanobeads coated with antibodies of both EpCAM and another mesenchymal marker Vimentin and employing lateral magnetophoresis to achieve high separation efficiency and reliability.

General comments:

Insufficient introduction of the application of CTCs and conventional tissue biopsy in cancer management. This knowledge background can facilitate the readers to understand the clinical significance and impact of the proposed research to current diagnostic and monitoring options of cancers, in particular, prostate cancer and pancreatic cancer.

Specific comments:

In Figure 5 (a), one CTC per milliliter was isolated in the blood samples of the healthy individual. Is this a false positive due to the method for CTC identification and enumeration (DAPI+CK+CD45-)? If it is so, would the false positive rate be proportional to the number of isolated CTCs?

Round 2

Reviewer 1 Report

In their resubmitted manuscript, Cha and colleagues have met most of my concerns, either by including more data of healthy volunteers or by amending their manuscript. The overall manuscript is now greatly improved and more concise.

Still, the ‘genetic proof’ of the cancerous origin of the cells detected in the very few clinical samples (n=5 for PDAC, n=17 for PC) reported on here remains missing, which is the major weakness of this otherwise nicely written and interesting manuscript.

As we all know, the detection of clinical CTCs from patient blood is – due to the unknown heterogeneity of the cells and differences between patients – magnitudes harder than the detection of clonal cell lines from TC spiked into healthy donor blood. Which is the reason why I still think this analysis is necessary to demonstrate that the isolation technique used here can really outperform ‘established’ methods. As the authors have not provided this information, the manuscript should be revised accordingly.

Even though this missing genetic information severely weakens the impact of this method, the manuscript should be published following the above minor revision. The technique is novel and the included experiments demonstrate the usability of the system at least in a wet-lab setting.
